# An Optical-Fiber-Based Key for Remote Authentication of Users and Optical Fiber Lines

**DOI:** 10.3390/s23146390

**Published:** 2023-07-14

**Authors:** Alexander Smirnov, Mikhail Yarovikov, Ekaterina Zhdanova, Alexander Gutor, Mikhail Vyatkin

**Affiliations:** Terra Quantum AG, Kornhausstrasse 25, CH-9000 St. Gallen, Switzerland; mj@terraquantum.swiss (M.Y.); ez@terraquantum.swiss (E.Z.); ag@terraquantum.swiss (A.G.); mvj@terraquantum.swiss (M.V.)

**Keywords:** optical fiber, optical communications, optical time domain reflectometry, physically unclonable function, Rayleigh backscattering, identification, authentication

## Abstract

We have shown the opportunity to use the unique inhomogeneities of the internal structure of an optical fiber waveguide for remote authentication of users or an optic fiber line. Optical time domain reflectometry (OTDR) is demonstrated to be applicable to observing unclonable backscattered signal patterns at distances of tens of kilometers. The physical nature of the detected patterns was explained, and their characteristic spatial periods were investigated. The patterns are due to the refractive index fluctuations of a standard telecommunication fiber. We have experimentally verified that the patterns are an example of a physically unclonable function (PUF). The uniqueness and reproducibility of the patterns have been demonstrated and an outline of authentication protocol has been proposed.

## 1. Introduction

Authentication is a practical approach for ensuring that data are transferred to the proper recipient [1]. One of the ways of authentication is the use of physically unclonable functions (PUFs) [2]. These are objects with a unique non-reproducible structure which evoke particular responses to different physical exposure. Researches have studied a vast variety of PUFs of different physical nature [3]. A significant part of PUFs is optical PUFs, in a majority of which interference plays the main role. Such PUFs may use biological layers [4], chemical solutions [5] or irregular 2D surfaces [6] as optical tokens. They also may utilize optical fibers as scattering media [7] or a waveguide [8]. However, the presence of interference severely limits the remote authentication properties of the conventional optical PUFs.

Recently, an experimental approach for identifying fiber sections based on Rayleigh scattering [9] analysis has been proposed [10,11,12,13,14,15,16]. This way of fiber section identification is not based on the optical interference and suggests using optical frequency domain reflectometry (OFDR) [17]. OFDR is characterized by a very high spatial resolution down to fractions of a millimeter. However, it does have distance limitations: it does not allow to look further than one-to-two kilometers, which is a severe disadvantage for remote measurement. Since fiber-optic lines are a common way of long-haul secure information transmission [18], the issue of remote optical fiber sections authentication is of great interest.

In this paper, we provided the physical model describing the mechanisms of the occurrence of the Rayleigh backscattering patterns and experimentally demonstrated that these patterns can be processed via Optical Time Domain Reflectometry (OTDR) [19]. Recent advances in the field of OTDR technology [20] make it promising for practical implementation in the fiber authentication. Although conventional OTDR devices have worse spatial resolution than OFDR, they allow measurements over significantly longer distances and are limited only by attenuation in the waveguide. Therefore, the more extensive distance range of OTDR devices is a major advantage for remote authentication. We considered the optical fiber sections in terms of PUF and interpreted OTDR-processed patterns obtained at distances of tens of kilometers as a “reflectokey” denotes the the unique signal that can be used for authentication. We experimentally checked the uniqueness and reproducibility of these patterns and suggested an outline of the authentication procedure. In addition, we investigated the spatial spectrum of OTDR-processed patterns and experimentally showed that they have a wide range of spatial frequencies, and could be detected on various spatial scales.

## 2. Method Description

The optical fiber span is a unique physical object itself. A preform for the fiber core is made of silica and doped with, e.g., Al, P, N, Ge, to give the fiber waveguide properties. Silica is an amorphous substance and every piece of it has a unique structure at the atomic level. Dopant atoms are randomly incorporated into the silica medium. All of these internal features are less than wavelength λ, and the light experiences Rayleigh scattering on them [9]. Additionally, the physical uniqueness of the optical fiber can be observed on much larger spatial scales, since it is also due to the specifics of the manufacturing of optical fiber. Furthermore, it can be detected via reflectometry methods such as OFDR and OTDR. This uniqueness provides the opportunity to obtain a distinctive backscattered signal, a “reflectokey”, which will authenticate any specific section of the fiber line.

The scheme of the experimental implementation of the OTDR approach [19] is shown in Figure 1. OTDR device sends light pulses of specific wavelengths, shapes and durations to probe the optical fiber line. In all our experiments, we fix a wavelength of λ = 1550 nm and apply probing pulses with selected duration, namely 200 ns, 500 ns or 1000 ns, and the repetition period of approximately 700 μs. The pulse sent generates a backscattered signal that characterizes the line. These subsequently averaged signals form a reflectogram representing the logarithmic dependence of the backscattered radiation intensity on the distance. Mathematically, a reflectogram is a convolution of a probing pulse and a particular function implying the unique individual extended characteristics of the optical fiber. In the next section, we provide a theoretical justification for this fact and prove that the reflectogram of the fiber line can be used to obtain authentication data, since it implicitly contains information about the physical uniqueness of the fiber.

In our experiments, we used the commercial OTDR device “Yokogawa AQ7275” which has the dynamic range of 32 dB for λ = 1550 nm and allows to measure the backscattered power level with a near $10^{-3}$ dB resolution. It is worth mentioning that its pulsed laser source was relatively broadband. The characteristic width of the spectral range of pulses in the vicinity of λ = 1550 nm was experimentally measured and turned out to be approximately 20 nm. As the fiber media, we used a standard single-mode telecommunication fiber “ITU-T G.652.D”. This fiber was in a spool. In all experiments, we used the same spool with a 50 km long fiber span. The physical connection of the fiber spool to the OTDR device was via a standard optical connector.

For homogeneous optical fibers, a linear decrease in the level of backscattered radiation, expressed in dB, with distance along the fiber is expected. However, in practice, it is easy to see that deviations from the linear decay take place (Figure 2a).

These fluctuations occur due to two factors. The first one is random noise, including the noise of the reflectometer components, e.g., the laser source or photodetector. However, the contribution of noise can be almost entirely eliminated by a proper averaging.

The second factor is a direct result of the unique inhomogeneities of the optical fiber. First of all, these are refractive index inhomogeneities, which are both local and non-local due to the technological features of fiber preform manufacturing. Moreover, additional technical inhomogeneities appear when manufacturing the optical fiber associated with the variability of the fiber geometry. Although special equipment controls the constancy of the core diameter when drawing the fiber from the preform, control accuracy exceeding a fraction of a percent is not a goal. As a result, even a tiny change in the core diameter may lead to a change in the resulting backscattering factor, which entails a difference in the level of the backscattered signal. All these effects contribute to the observed fluctuations in the reflectogram. This contribution determines the variations shown in Figure 2a. Furthermore, it is this contribution that we propose to use as a unique key for the corresponding fiber section—its “fingerprint”.

These unique patterns can be observed, e.g., by subtracting the linear contribution from the reflectogram. Figure 2b shows an example of the processing result for a 1 km section of standard SMF-28e (ITU-T G.652.D) single-mode telecommunication fiber. The measurements were carried out with an OTDR approach at a wavelength of λ = 1550 nm with a probing pulse duration of 500 ns and averaging over $2^{10}$ pulses, which is enough for the elimination of random noises.

It can be experimentally verified that in successive measurements with the same pulse parameters, the patterns for the same fiber section are reproduced with high accuracy (the blue and red curves in Figure 2b are almost entirely coincide). As it will be shown below, the fundamental properties of the backscattered responses will be characterized by calculating inter- and intra- $L_{2}$ distances [2] between patterns showing identifiability and robustness of the approach.

## 3. Results

### 3.1. Theoretical Analysis

The source of the observed patterns is the variations of backscattered Rayleigh radiation. It is of interest how the backscattered light implies information about the internal properties of fiber media. The signal is formed by local scattering acts, which are determined by local properties of the fiber: the refractive index, attenuation constant and backscattering factor, which may vary with distance. These value variations over fiber length should be described and connected with detected power. Following the standard approach [21,22], we derive these variations.

To obtain the backscattered power from a local fiber section, let us consider electric field E(x,y,z,t) of the initial linearly polarized speculative electric pulse propagating in the fiber in the *z* direction as
(1)$$E\left(x,y,z,t\right)=E_{0}\hat{\psi }\left(x,y\right)f\left(z-v_{g}t\right)\exp\left(i\beta \left(\omega \right)z-i\omega t\right)\exp\left(-\frac{1}{2}\int_{0}^{z}\alpha \left(\xi ,\omega \right)d\xi \right),$$
where $E_{0}$ denotes the pulse’s magnitude, $\hat{\psi }\left(x,y\right)$ denotes the field distribution of the fundamental mode, $v_{g}$ denotes the group speed of light, β denotes the propagation constant, ω denotes the light frequency and α denotes the attenuation constant. $f(z-v_{g}t)$ is a rectangular pulse envelope function with width *l*:(2)$$f\left(x\right)=\left\{\begin{array}{cc} 1, & \;\mathrm{if}\;x\in [-l/2,l/2]\\ 0, & \;\mathrm{else}\; \end{array}\right..$$

The usual formalism used in optical communication theory expresses the pulse shape as a function of exclusively time. Here, the travelling wave formalism for the pulse shape was chosen in order to describe the interaction between the fiber medium and the pulse.

The differential amplitude (at the moment *t* in z=0) of the electric field scattered from section $dz_{s}$ at point $z=z_{s}$:(3)$$\begin{array}{c} dE_{s}\left(x,y,z_{s},t\right)=dz_{s}\frac{E_{0}}{\pi w_{0}^{2}}\frac{\omega }{nc}\hat{\psi }\left(x,y\right)f\left(2z_{s}-v_{g}t\right)\exp\left(-z_{s}\bar{\alpha }\left(z_{s}\right)\right)\exp\left(2i\beta z_{s}-i\omega t+\pi /2\right)\times \\ \times \left(\int_{S}dxdy\Delta \chi \left(x,y,z_{s}\right){\left|\hat{\psi }\right|}^{2}\left(x,y\right)\right), \end{array}$$
where $\Delta \chi (x,y,z_{s})$ denotes the local inhomogeneities of electric susceptibility which cause the scattering, $\bar{\alpha }\left(z_{s}\right)$ denotes averaged value of α(z) over the distance $\left[0;z_{s}\right]$, $w_{0}$ denotes the mode field diameter while the field distribution is assumed to have a Gaussian form: $\hat{\psi }\left(x,y\right)=\exp\left(-\left(x^{2}+y^{2}\right)/w_{0}^{2}\right)$. Mode field distribution, in general, may slightly vary on *z*, but this dependence is insignificant since these fluctuations are averaged.

Integration over the pulse length gives the following:(4)$$\begin{array}{c} E_{s}\left(x,y,0,t\right)=\int_{v_{g}t/2-l/4}^{v_{g}t/2+l/4}dE_{s}\left(x,y,z_{s},t\right). \end{array}$$

Now, we assume that the exponential decay $\exp(-z_{s}\bar{\alpha }\left(z_{s}\right))$ does not change on the scales of the wavelength. That is true for the typical attenuation constant α which is about 0.18–0.24 dB/km. Moreover, this is also justified by the fact that α makes sense only on the scales of several wavelengths. This makes it possible to calculate backscattered power:(5)$$\begin{array}{c} P_{s}\left(t\right)=\int_{S}dxdy\frac{|E_{s}{|}^{2}\left(t\right)}{2}\sqrt{\mu /\epsilon }=P_{0}{\left(\frac{\omega }{c}\right)}^{2}e^{-\bar{\alpha }v_{g}t}\frac{Q(t)}{n^{2}{\left(\pi w_{0}^{2}\right)}^{2}}, \end{array}$$
where the initial power is $P_{0}=\left(1/2\right)\left(E_{0}^{2}/\sqrt{\mu /\epsilon }\right)\left(\pi w_{0}^{2}/2\right)$ and
(6)$$\begin{array}{c} Q\left(t\right)={\left|\int_{v_{g}t/2-l/4}^{v_{g}t/2+l/4}dz_{s}e^{2i\beta z_{s}}\int_{S}dxdy\Delta \chi \left(x,y,z_{s}\right){\left|\hat{\psi }\right|}^{2}\left(x,y\right)\right|}^{2}. \end{array}$$

For further simplicity, we denote the probing pulse center coordinate by $z_{p}=v_{g}t/2$. The backscattered power at the input of the fiber at the moment *t* corresponds to backscattering from the section of l/2 length at the point $z_{p}$. Assuming that *l* is at least several times larger than both atomic scales and wavelength, we rewrite Q(t) as
(7)$$Q\left(z_{p}\right)={\left|{\hat{\chi }}_{2\beta }\left(z_{p}\right)\right|}^{2}={\left|\int_{z_{p}-l/4}^{z_{p}+l/4}dz_{s}e^{2i\beta z_{s}}\bar{\Delta \chi }{\left(z_{s}\right)}_{S}\right|}^{2},$$
where $\bar{\Delta \chi }{\left(z_{s}\right)}_{S}$ denotes a weighted average of $\Delta \chi (x,y,z_{s})$ with the squared distribution of the fundamental mode $|\hat{\psi }{|}^{2}\left(x,y\right)$. $Q(z_{p})$ in Equation (Equation 7) can be treated as the squared absolute value of spatial Fourier transform of electric susceptibility with spatial vector 2β of the fiber section in the vicinity of the point $z_{p}$. Equation (Equation 5) can be rewritten as
(8)$$P_{s}\left(z_{p}\right)=\frac{P_{0}}{n^{2}{\left(\pi w_{0}^{2}\right)}^{2}}e^{-2z_{p}\bar{\alpha }}{\left(\frac{\omega }{c}\right)}^{2}{\left|{\hat{\chi }}_{2\beta }\left(z_{p}\right)\right|}^{2}.$$

Equations (Equation 7) and (Equation 8) completely describe the scattering process in single-mode optical fibers, showing how the reflected light brings the information about internal structure of the fiber. The power scattered in the vicinity of the auxiliary point of the fiber is defined by the local electric susceptibility distribution, refraction coefficient and mode field distribution. The mode field distribution may also vary in different fiber sections due to geometry defects and refractive index variability. For convenience, $z_{p}$ can be replaced with *z*. Following the results obtained in previous works [21,22], $|{\hat{\chi }}_{2\beta }{\left(z\right)|}^{2}$ can be linked with observable value α(z):(9)$$\alpha \left(z\right)=\frac{4{\left(\omega /c\right)}^{4}}{3\pi^{2}w_{0}^{2}\left(z\right)l}{\left|{\hat{\chi }}_{2\beta }\left(z\right)\right|}^{2}.$$

In Equation (Equation 9), α(z) is the coefficient of loss due to the scattering. Similar to $|{\hat{\chi }}_{2\beta }{\left(z\right)|}^{2}$ in Equation (Equation 7), α in Equation (Equation 9) can be defined only at a fiber section the size of at least several wavelengths. Otherwise, it loses its physical meaning. Preferably, this fiber section length l/2 should be more than several dozens of wavelengths. Together with the backscattering factor, it determines the power scattered in the backward direction. Thus, from Equation (Equation 9), we can see that apart from the refractive index and the mode field diameter, possible variations of the backscattered radiation are due to possible oscillations of $|{\hat{\chi }}_{2\beta }{|}^{2}$ along the fiber.

To derive the power of the real pulse backscattered from the length l/2, let us substitute Equation (Equation 9) into Equation (Equation 8) and obtain:(10)$$P_{s}\left(z\right)=P_{0}\frac{l}{2}\alpha \left(z\right)B\left(z\right)e^{-2z\bar{\alpha }}=W\frac{v_{g}}{2}\alpha \left(z\right)B\left(z\right)e^{-2z\bar{\alpha }}.$$

Here, $B\left(z\right)=\frac{3}{2n{\left(z\right)}^{2}w_{0}{\left(z\right)}^{2}}{\left(\frac{c}{\omega }\right)}^{2}$ denotes the backscattering factor with n(z) and $w_{0}\left(z\right)$ depending on distance in general case. $W=P_{0}\tau =\frac{P_{0}l}{v_{g}}$ denotes a pulse energy.

$P_{s}\left(z\right)$ from Equation (Equation 10) may be considered as backscattered power from a delta-like probing pulse. This assumption is justified for a narrow pulse. As discussed above, the pulse length should be no less than several wavelengths. This length should be more than the coherence length to avoid interference effects. In consideration of this, α(z) can be treated as value at the point.

*W* corresponding to delta-pulses should be replaced with $W\times F(2z-v_{g}t)dz$ for the general case of extended pulses. Here, a normalized power envelope function $F(z-v_{g}t)$ describes the pulse shape. The backscattered power for extended pulses may be expressed as
(11)$$P_{s}\left(t\right)=\frac{Wv_{g}}{2}\int_{0}^{L}dz\alpha \left(z\right)B\left(z\right)F\left(2z-v_{g}t\right)e^{-2z\bar{\alpha }},$$
where *L* denotes the total length of the fiber. Thus, the backscattered power is a convolution of the pulse with α(z) and B(z), which describe the unique scattering properties of the optical fiber, e.g., for the rectangular optical pulses, which are relatively short compared to an exponential decay, Equation (Equation 11) takes the form of a window-averaged product of α and *B*:(12)$$P_{s}\left(t\right)=\frac{Wv_{g}}{2}{\bar{\alpha (z)B(z)}}_{\left[z-\delta /4,z+\delta /4\right]}e^{-2z\bar{\alpha }},$$
where δ corresponds to the pulse length. This equation demonstrates that individual scattering properties may be observed through conventional OTDR technology where the typical pulse duration is ∼1 ns–10 μs (∼0.2 m–2 km).

Rayleigh scattering is due to fluctuations of $\Delta \chi (x,y,z_{s})$ on the scales less than optical signal wavelength. These small-scale fluctuations, often regarded as white spatial noise, determine the value of $|{\hat{\chi }}_{2\beta }{|}^{2}$, which causes the optical signal backscattering. Otherwise, the inhomogeneities in optical fibers are in an extensive range of spatial scales. Therefore, they will define variations of $|{\hat{\chi }}_{2\beta }{|}^{2}$ on scales larger than the optical signal wavelength. As these $|{\hat{\chi }}_{2\beta }{|}^{2}$ variations in an optical fiber may have an extensive spatial range on the larger scales, they can also be described, at first approximation, as the white spatial noise. These large-scale variations can be observed with, e.g., a conventional OTDR technique.

### 3.2. Experimental Verification of Fiber Key Uniqueness and Reproducibility

As mentioned above, variations remaining on the reflectogram (i.e., patterns) are due to different chaotically distributed inhomogeneities of the fiber waveguide. Furthermore, since it is technologically impossible to reproduce the exact locations of such inhomogeneities, the fiber sections can be considered in terms of PUF in the language of challenge–response pairs [2]. As a result, they, as physically unique objects, can be considered as keys for authentication after conducting reflectometric measurements (in our case, OTDR measurements) with light pulses having predetermined parameters.

As was stated earlier, the OTDR technique may demonstrate several kinds of instability, particularly in laser pulse intensity or shape impermanence, photodetector noises, delay mismatching, etc. This instability may not be eliminated sufficiently by proper averaging to achieve the required properties of uniqueness and reproducibility. Thus, these properties should be clearly verified experimentally.

The successive measurements were carried out with the same pulse parameters for two independent 400 m long sections of the same fiber from the 50 km long fiber span in the spool. Namely, the first section was 20.0–20.4 km from the OTDR device and the second was 20.5–20.9 km. Five consecutive measurements of the span were made, and after that, patterns corresponding to each fiber section were obtained by subtracting the linear contribution from the reflectograms. Based on the obtained patterns, the joint correlation matrix was constructed. This matrix is presented in Figure 3a. The patterns of fiber sections are individually well reproduced (red blocks), as the values of the standard correlation coefficient are not less than 0.93. On the other hand, patterns of independent sections of the fiber are entirely different (blue blocks), since the absolute values of the correlation coefficient do not exceed 0.26. One can see that even sections of the same fiber may be distinguished with a high degree of accuracy.

Backscattered power is conjointly determined by the probing pulse and fiber waveguide properties. Therefore, if the parameters of the pulse vary, the resulting patterns may also change. An increase in the duration of pulses leads to deterioration in the effective spatial resolution of the OTDR device, thereby eliminating the variations with high spatial frequencies and changing the appearance of patterns. Thus, the same fiber affected by pulses with distinct shapes should produce essentially different patterns. To experimentally confirm this fact, we measured a fixed section of the fiber, applying probing pulses of various duration: 500 ns and 1000 ns. In Figure 3b, a similar matrix is shown. Again, the section which is 20.0–20.4 km from the OTDR device was chosen. The patterns for each series of individual measurements are well reproduced again since the correlation coefficient values are at least 0.90. Still, when the pulse parameters change, the patterns become dissimilar.

To prove the uniqueness and reproducibility of patterns, the statistics were obtained for the inter- and intra- distances between them. Figure 3c shows the results of the tests. In total, 104 different 1 km long fiber sections at various distances up to 27 km from the reflectometer were processed, including even significantly remote fiber sections with worse reproducibility due to higher relative noise. The statistical distribution for the $L_{2}$ distance between successive measurements of these sections corresponding to 5700 pairs of patterns is presented by the pink histogram. This distribution demonstrates the Poisson-like probability with a mean value of 0.11.

Statistics for inter-distance corresponding to 137,500 pairs of patterns are presented by the blue histogram. These statistics demonstrate the Gaussian distribution with a mean value of 0.44 and a variance of $5.6\times 10^{-3}$. The conducted tests show that the overlap of histograms is insignificant. As a result, the distance threshold appropriate for the successful 1 km long fiber section identification can be chosen as 0.25.

According to Equation (Equation 12), the backscattering signal value is defined via window averaged product of α and B on the pulse length. It means that the quality of reproducibility and uniqueness depends on the fiber section and the pulse length ratio. In the above experiments, this ratio value was 10, which seems close to the lower limit. However, even for this case, the uniqueness and reproducibility were established at a fairly high level.

Additionally, the stability of correlation coefficient values over time was checked. The patterns were again processed for the same 400 m long fiber section, which is 20 km away from the reflectometer. Twenty five measurements were sequentially made on one day and the same number of measurements a week later. Figure 4 shows the corresponding joint correlation matrix. All matrix elements are greater than 0.80, which indicates good reproducibility of patterns with time. It is also worth noting that an increase in the fiber section length leads to additional reproducibility improvement.

The obtained patterns indeed demonstrate uniqueness and reproducibility, which are required for identification purposes. Therefore, our experimental results prove that an optical fiber section may be recognized in the authentication procedure.

### 3.3. Spatial Frequencies Spectrum Investigation

On the one hand, we experimentally observed unique patterns on the scales of tens of kilometers with the resolution from tens to hundreds of meters. On the other hand, patterns may also be observed with the OFDR technique on the scales of meters with submillimeter resolution [10,11,12,13]. These patterns correspond to independent variations of the fiber media. As unique patterns can be observed in a wide range of spatial scales, it is of interest to investigate their spatial spectrum.

To determine characteristic spatial frequencies of observed variations of patterns, the discrete Fourier transform (DFT) of patterns has been carried out (Figure 5a). For convenience, the spatial vector k was used, related to the spatial period Λ as $k=\frac{2\pi }{\Lambda }$. The patterns were obtained in OTDR measurements of the first 25 km long fiber section of 50 km long fiber span in the spool with pulse durations of 200 ns and 1000 ns (Figure 5b).

According to the results, the spatial spectrum is dense with frequencies from ∼$10^{-4}$ m${}^{-1}$ to ∼$10^{-1}$ m${}^{-1}$. Moreover, for both pulse durations, the amplitudes of the lowest spatial frequencies have essentially non-zero values (Figure 5a, subplot). Thus, spatial periods of variations of patterns may reach scales of the length of fiber line and are limited only by OTDR distance range or the fiber length.

In a high spatial frequency region, the spectrum is limited by the pulse length, which determines the spatial resolution of a device. As an estimate of the boundary for high spatial frequencies, we considered the point at which the Fourier transform value is halved compared to the maximum. For convenience, a polynomial approximation was applied to the data (dashed lines in Figure 5a). For the selected pulses, these boundaries correspond to a spatial period of about fifty meters for a pulse duration of 200 ns and about two hundred meters for a pulse duration of 1000 ns. These values are of the order of corresponding pulse lengths in the fiber waveguide which are $l=v_{g}\times \tau_{p}$, where $v_{g}\approx 2\times 10^{8}$ m/s denotes the speed of light in optical fiber, and $\tau_{p}$ denotes the duration of the pulses.

Figure 5a also includes the spatial spectrum of the OTDR device’s probing pulse. The length of the pulse is 200 m which corresponds to its duration of 1000 ns. It can be seen that this spectrum looks like the envelope function for corresponding spatial frequency spectrum. That is also in agreement with the theoretical predictions. In accordance with Equation (Equation 11), since the backscattered signal is a convolution of the probing pulse shape with the scattering function of the fiber, the resulting spatial spectrum should be their product in Fourier space: $F\left[f\left(x\right)*g\left(x\right)\right]=\hat{f}\left(k\right)\hat{g}\left(k\right)$. Wherein, zero values of the spatial spectrum should coincide with corresponding zero values of the rectangular pulse spectrum, what is indeed observed. Moreover, since the pulse spectrum value is substantially constant in the vicinity of k=0, the measured spatial spectra in this range have fairly close values for different pulse durations (Figure 5a, subplot).

The above results indicate that OTDR-processed pattern variations can be observed in a wide range of spatial periods from tens of meters to tens of kilometers. Considering the results presented in OFDR experiments, the overall spectrum extends from submillimeter scales to tens of kilometers and possibly even greater values. Since the spectrum is so broad and dense, the patterns’ oscillations’ structure is similar to that of white noise. Furthermore, the observations mentioned above provide extremely high robustness against altering the fiber section by the intruder. These results additionally confirm the possibility of using these patterns to authenticate remote users or fiber sections of various lengths.

## 4. Discussion

The uniqueness of the patterns obtained from OTDR measurements can be used for the authentication of important sections of a fiber line or almost the entire line. As a possible practical algorithm, the following sequence of actions is suggested for the user of the fiber line:The user first carries out OTDR measurements with a large variety of available challenge pulses to form a challenge–response pair database in digitized form.To authenticate an optical line, a legitimate user needs to carry out OTDR measurements with randomly selected challenge pulse(s) from their database and collect response(s).The user then takes an appropriate mathematical comparison method (e.g., correlation metrics, Hamming distance, $L_{2}$ distance) for their response(s) and obtains the result of whether the authentication is successful.

The individual sample of fiber can be used to authenticate the particular legitimate user. If it is necessary to exchange information between two users, they will first connect their fiber samples, each from its side, and authenticate each other in turn. The physical connection can be implemented, e.g., by employing optical connectors or fusion splices.

We propose the sketch of the following authentication protocol on the example of a two-user model (Alice and Bob) without a detailed analysis of cryptographic strength:Both Alice and Bob, with their OTDR devices, carry out in advance numerous measurements of each other’s unique fiber samples with a wide variety of probing pulse parameters, i.e., form two databases of challenge–response pairs [2].For Alice to authenticate Bob, he connects their unique fiber sample to the receptacle. Then, Alice randomly selects challenge–response pair(s) from her database and sends the light pulses with corresponding parameters to Bob.In the case of correct response in terms of the chosen method of mathematical comparison, Bob authenticates Alice the same way.In the case of correct responses from both sides, authentication is successful.

It is worth emphasizing that the considered variations of the backscattered signal level are in no way connected with interference effects, on which many optical PUFs are based [3]. The explanation of the interference absence is that the coherence length of the OTDR device laser source is smaller than the length of the probing pulses [23], namely, $l_{c}\ll l_{p}$. These lengths can be estimated as $l_{c}∼\frac{\lambda^{2}}{\Delta \lambda }≲10^{-4}$ m and $l_{p}∼c\times \tau_{p}≳10$ m, respectively, where $\Delta \lambda ∼10^{-8}$ m denotes the the width of the spectral range of pulses of the OTDR device used in our experiments, $c=2\times 10^{8}$ m/s denotes the speed of light in optical fiber waveguide, and $\tau_{p}≳100$ ns denotes the the duration of the applied probing pulses. The absence of interference gives an advantage over conventional interference-based optical PUFs due to the patterns’ stability in time and remote control availability. If interference effects were present, we would not be able to observe the reproducibility of patterns after a long time because of temperature and mechanical vibration effects. As the time between the measurements was significant, a difference in the ambient temperature and, as a result, in the fiber temperature inevitably occurred. Furthermore, since the refraction coefficient of the optical fiber and fiber length significantly depend on temperature, the optical paths’ lengths also change with temperature. The alteration of the latter would essentially vary the interference results.

However, some decrease in correlations between patterns over the week was still found. We believe this is due to the OTDR device instabilities, namely, the laser pulse intensity variations or its shape impermanence. So, it is of interest to create an in-house setup with more stabilized laser source and investigate the long-term stability of the pattern’s reproducibility.

## 5. Conclusions

We demonstrated a conventional OTDR technology as a convenient instrument for investigating unique variations of backscattered Rayleigh radiation in optical fiber waveguides. The patterns obtained from reflectograms of the fiber sections were detected in a wide range of spatial scales from tens of meters to tens of kilometers.

We have developed a theoretical model which explains the physical nature of the backscattered radiation patterns and showed that the physical origin of the observed patterns is a convolution of optical pulse with a unique scattering profile of the fiber. The fiber manufacturing process explicitly implies the presence of extended inhomogeneities, which allow the observation of patterns on an even broader scale from submillimeters to hundreds of kilometers. The observed spatial oscillations spectra are broad and dense in three orders of magnitude range.

The conducted experiments along with the provided model make it possible to perform remote authentication even with a worse spatial resolution compared to the OFDR technique. This fact gives the opportunity to observe the unique inhomogeneities of the internal structure of an optical fiber waveguide by the OTDR technique using relatively long pulses, with durations in the order of microseconds. Long OTDR pulses present the opportunity to obtain the backscattered signal from the remote fiber sections.

We examined a large set of statistical data and experimentally verified the uniqueness and reproducibility of the patterns using inter- and intra-distance distribution with a $L_{2}$-norm. In addition, the sketches of the authentication procedures for the long-haul optical fiber line sections or legitimate remote users were proposed. In our forthcoming publications, we are going to analyze the presented approach from a cryptographic point of view and consider the implementation of the concept in common data transmission systems.

Future work in this area may also study the effect of external physical conditions on the reproducibility of backscattering patterns. It is also of interest to involve more advanced mathematical methods in evaluating the uniqueness of patterns, particularly machine learning and neural network technologies.

## Figures and Tables

**Figure 1 sensors-23-06390-f001:**
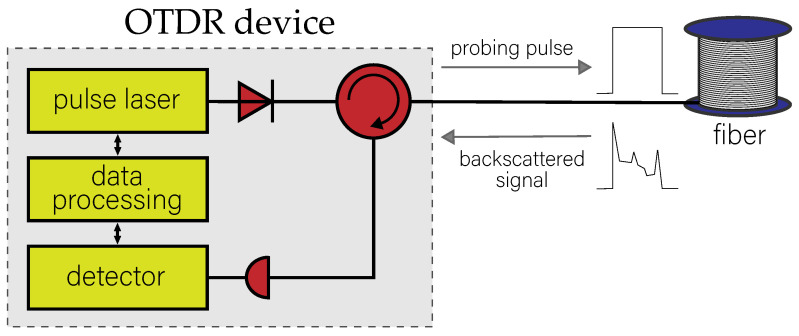
The schematic description of the optical time domain reflectometry (OTDR). The internal structure of the OTDR device is illustrated schematically.

**Figure 2 sensors-23-06390-f002:**
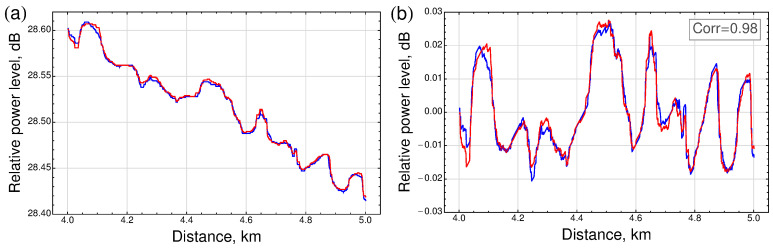
(**a**) An example of experimentally obtained reflectograms corresponding to successive measurements of a homogeneous 50 km long fiber span in a spool. The section corresponding to the fifth kilometer is shown. The red curve corresponds to a measurement taken about 10 s after the measurement, which corresponds to the blue curve. Deviations from the linear trend are mainly due to inhomogeneities in the optical fiber. Both measurements were carried out for a standard single-mode fiber SMF-28e (ITU-T G.652.D) at a wavelength of λ = 1550 nm with a probing pulse duration of 500 ns and averaging over $2^{10}$ pulses. (**b**) An example of patterns remained after subtracting the linear contribution from the reflectograms shown in (**a**). High-precision reproducibility of patterns with a correlation coefficient value of 0.98 is observed.

**Figure 3 sensors-23-06390-f003:**
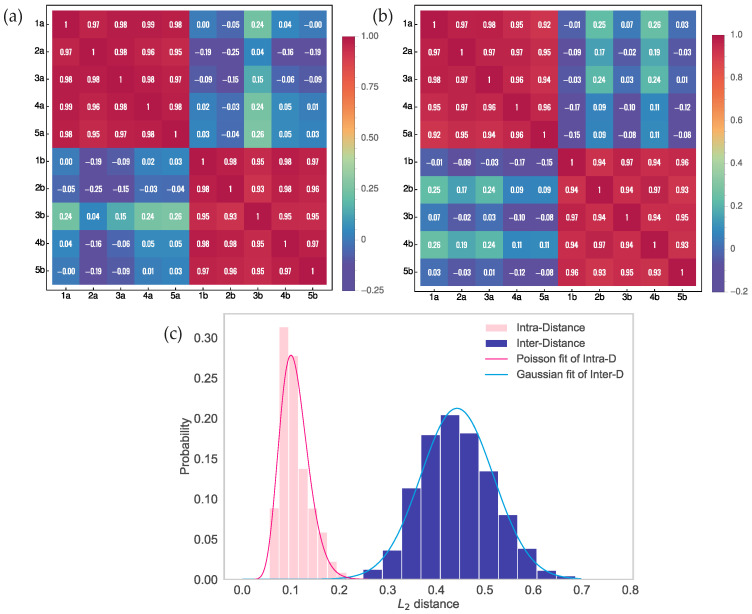
Experimental confirmation of uniqueness of fiber keys. (**a**) Joint correlation matrix for two series of 5 measurements of two sections of a standard SMF-28e (ITU-T G.652.D) single-mode fiber, each 400 m long. The nearest section is 20 km away from the reflectometer. The measurements were carried out at a wavelength of λ = 1550 nm with a pulse duration of 500 ns and an averaging of $2^{10}$. Series “a” corresponds to the first fiber section and series “b” corresponds to the second. (**b**) Joint correlation matrix for two series of 5 measurements of single fiber section 400 m long, which is 20 km away from the reflectometer. The measurements were carried out at a wavelength of λ = 1550 nm with $2^{10}$ averaging, but with various pulse duration. Series “a” corresponds to the duration of 500 ns and series “b” corresponds to the duration of 1000 ns. (**c**) Experimentally obtained statistical distributions of the reflectokey inter- and intra- distances with $L_{2}$ norm. The blue and pink histograms correspond to 137,500 and 5720 pairs of patterns of the different and same 1 km long fiber sections, accordingly. The compared fiber sections were considered at distances up to 27 km. The measurements were carried out at a wavelength of λ = 1550 nm with a pulse duration of 500 ns and an averaging of $2^{13}$.

**Figure 4 sensors-23-06390-f004:**
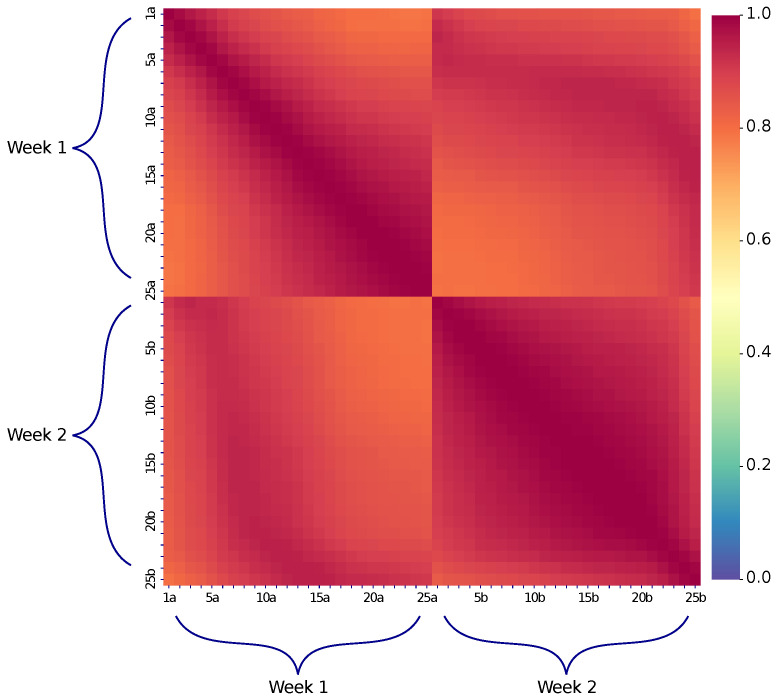
A joint correlation matrix for two measurement series of a section of a standard SMF-28e (ITU-T G.652.D) single-mode fiber. The measurements were carried out at a wavelength of λ = 1550 nm with a pulse duration of 500 ns and an averaging of $2^{14}$. The time difference between consecutive experiments into same series is 2 min. The time difference between the series is one week. Since all values of the correlation matrix are greater than 0.80, long-time reproducibility of observed patterns is established.

**Figure 5 sensors-23-06390-f005:**
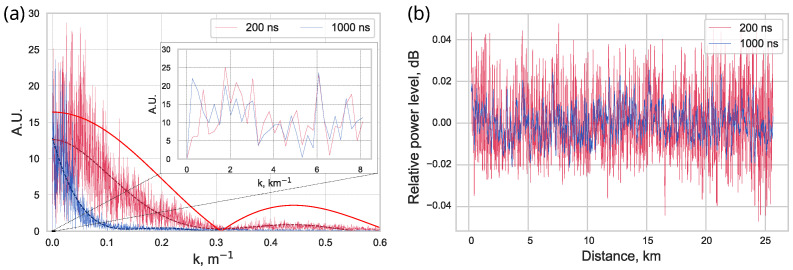
(**a**) The absolute values of discrete Fourier transform (DFT) of signal patterns represented in (**b**). Dashed lines represent the polynomial approximations of data. The subplot represents the starting region of the main plot. Solid line represents the spatial spectrum of the probing pulse sent by the OTDR device. (**b**) Patterns corresponding to standard SMF-28e (ITU-T G.652.D) single-mode fiber section, which is 0–25 km away from the reflectometer. The red curve corresponds to a measurement with a pulse duration of 200 ns and the blue curve corresponds to a measurement with a pulse duration of 1000 ns. Both measurements were carried out at a wavelength of λ = 1550 nm with an averaging of $2^{10}$.

## Data Availability

All materials that support the results of this study are available from the corresponding author upon reasonable request.

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
