# Peer review of "An Optical-Fiber-Based Key for Remote Authentication of Users and Optical Fiber Lines"

_sensors, 2023, doi:10.3390/s23146390_

Round 1
Reviewer 1 Report
Please, see the referee report.

Reviewer 2 Report
In the manuscript under reviewing, an optical fiber-based key for remote authentication of users and optical fiber line is proposed and verified theoretically and experimentally. The technical problem addressed in the manuscript is the opportunity to use a conventional Optical Time Domain Reflectometry (OTDR) technology as a convenient instrument for remote authentication of unique variations of backscattered Rayleigh radiation in optical fiber waveguides. To realize it, the authors developed a theoretical model, which explains the physical nature of the backscattered radiation patterns, and showed that the physical origin of the observed patterns is a convolution of optical pulse with a unique scattering profile of the fiber. The conducted experiments along with the provided model showed the possibility to perform remote authentication even with a worse spatial resolution compared to conventional Optical Frequency Domain Reflectometry (OFDR) technique. In the result, the opportunity to observation of the unique inhomogeneities of the internal structure of an optical fiber waveguide by OTDR technique using relatively long pulses, with duration of the order of microsecond. I believe that this article has all the necessary properties in terms of novelty of the solution and readability to be accepted for publication. I would like to especially note the clarity and detail of the spatial frequencies spectrum investigation (see sub-section 3.3). Nevertheless, I believe that its readability will improve significantly if the model and key characteristics of the applied optical reflectometer (the resolution for measuring the reflected power level, etc.) are indicated in the course of the experimental study (see Figs. 2, 5(b)), since I am sure that readers who are familiar with measurements of scattering in optical fiber, just like me, the accuracy of measurement at the level of 0.01 dB is very alarming. I also think that it is worth specifying the repetition period of the probing optical pulses used in the measurements. In addition, I note that the “Power level” (see Figs. 2 and 5(b)) is measured in dBm, and in dB - the “Relative power level”. Following these, I recommend accepting the manuscript with minor revision.
Reviewer 3 Report
The authors have proposed physical model describing the mechanisms of the occurrence of the Rayleigh backscattering patterns and experimentally demonstrated that these patterns can be processed via Optical Time Domain Reflectometry (OTDR) for remote authentication. I believe that the work done and results are trivial. No significant achievement has been carried out. The similar research has already been published. For example
· Goki, Pantea Nadimi, Stella Civelli, Emanuele Parente, Roberto Caldelli, Thomas Teferi Mulugeta, Nicola Sambo, Marco Secondini, and Luca PotÌ. "Optical identification using physical unclonable functions." arXiv preprint arXiv:2305.02141 (2023).
· Yao, Zheyi, Thomas Mauldin, Gerald Hefferman, Zheyu Xu, Ming Liu, and Tao Wei. "Low-cost optical fiber physical unclonable function reader based on a digitally integrated semiconductor LiDAR." Applied Optics 58, no. 23 (2019): 6211-6216.
The backscattered light from a discontinuity can be used for different purposes, as well known in literature. So no new in it. Moreover, the work done seems not experimental. It is some sort of numerical analysis. No test bed, equipment used, manufacturer, model etc is described. Therefore, I recommend its rejection.

NIL.
Round 2
Reviewer 3 Report
I have gone through the revised version and authors response. I hold my previous decision that this paper has nothing new in it. The back scattered light can be used for different purposes including remote authentication, that has been widely studied in published works. This work is extension of those past works that I already have mentioned. Therefore, I am of the view that this paper should not be published in high impact factor journal like Sensors. I recommend its rejection.
